# All-optical nonreciprocity due to valley polarization pumping in transition metal dichalcogenides

Sriram Guddala[1], Yuma Kawaguchi[1], Filipp Komissarenko[1], Svetlana Kiriushechkina[1], Anton Vakulenko[1], Kai Chen[1,2], Andrea Alù [1,2,3], Vinod M. Menon [2,4] & Alexander B. Khanikaev [1,2,4 ✉]

Nonreciprocity and nonreciprocal optical devices play a vital role in modern photonic technologies by enforcing one-way propagation of light. Here, we demonstrate an all-optical approach to nonreciprocity based on valley-selective response in transition metal dichalcogenides (TMDs). This approach overcomes the limitations of magnetic materials and it does not require an external magnetic field. We provide experimental evidence of photoinduced nonreciprocity in a monolayer $WS_2$ pumped by circularly polarized (CP) light. Nonreciprocity stems from valley-selective exciton population, giving rise to nonlinear circular dichroism controlled by CP pump fields. Our experimental results reveal a significant effect even at room temperature, despite considerable intervalley-scattering, showing promising potential for practical applications in magnetic-free nonreciprocal platforms. As an example, here we propose a device scheme to realize an optical isolator based on a pass-through silicon nitride (SiN) ring resonator integrating the optically biased TMD monolayer.

[1] Department of Electrical Engineering, Grove School of Engineering, City College of the City University of New York, New York, NY, USA. [2] Physics Program, Graduate Center of the City University of New York, New York, NY, USA. [3] Photonics Initiative, Advanced Science Research Center, City University of New York, New York, NY, USA. [4] Department of Physics, City College of New York, New York, NY, USA. ✉email: akhanikaev@ccny.cuny.edu

Nonreciprocal optical devices, such as isolators and circulators, are critical components for photonic systems[1–9] at large. Optical isolators enable stable laser operation by blocking reflected light from entering the laser cavity, and circulators facilitate nonreciprocal routing of optical signals in telecommunication networks. However, nonreciprocal devices available today rely on magneto-optical materials, which have limited possibility of integration into modern photonic circuitry due to chemical incompatibility of materials and typically require a bulky external magnetic bias. In addition, the weak character of magneto-optical effects prevents miniaturization of magneto-optical components, which must be large to provide sufficient nonreciprocal response. Although numerous solutions have been proposed, including photonic[7,10–14] and plasmonic nanomaterials and microstructures[15–20] integrating magneto-optical media, these schemes have yet to be proven of technological relevance.

In recent years, magnet-free approaches to nonreciprocity have gained attention, including linear[14]- and angular-[21] momentum biased photonic structures and metamaterials. In such systems, parametric phenomena induced by external time-modulated bias were shown to give rise to nonreciprocal responses. However, electro-optical modulation schemes are limited to a few GHz speeds, implying that optical nonreciprocity can be difficultly achieved with these schemes, and most experimental demonstrations with practically relevant metrics of performance have been limited to radio-frequencies[9,22]. Nonlinear phenomena combined with asymmetric field distributions have also been shown to enable nonreciprocity in some regimes, exploiting the temporal modulations enabled by the signal itself as it propagates through the device[23,24]. However, this form of self-bias nonreciprocity comes with some drawbacks[24–27], such as intensity-dependent operation, signal distortion, trade-offs between insertion loss and bandwidth, as well as the requirement to be operated in pulsed regime, which overall hinder its widespread applicability. Some of the all-optical modulation schemes to break reciprocity proposed recently offer new approaches to nonreciprocity[28] based on optical nonlinearities. Nonetheless, in the optical domain magnet-free isolators and circulators remain elusive, even though some important proof of concept experimental schemes have been demonstrated[7,8].

In a different context, two-dimensional (2D) Van der Waals materials have been shown to provide a promising platform for enhanced light–matter interactions, including enhanced nonlinear responses in graphene and other 2D materials[29–34]. A particular class of 2D semiconductors, monolayer transition metal dichalcogenides (TMDs), has attracted significant attention from the research community due to their unique valley-dependent optical response[35–38]. The conservation of angular momentum in TMDs enforces circularly polarized (CP) light to interact selectively with electronic subsystems at K and K′ valleys, leading to valley-selective absorption of CP light[39,40]. Valley-polarized excitons have been shown to support CP luminescence connected with the pump handedness, due to the conservation of the valley degree of freedom[41–45]. A variety of fascinating effects based on such chiral light-matter interactions have been demonstrated recently, including directional launching of guided waves with TMD monolayers integrated into photonic topological insulators[46,47] and surface plasmon-polaritons[48–51]. More recently nonlinear effects in TMDs such as saturable absorption[52], valley-dependent exciton bistability[53], and valley-dependent second harmonic generation[54–56] have been demonstrated and proposed for valley optoelectronics applications.

In this work, we exploit the valley selective response of TMDs to experimentally demonstrate that chiral light–matter interactions in these materials open a route to all-optical nonreciprocal photonics. We show that the valley-selective exciton population leads to photoinduced nonreciprocal circular dichroism analogous to the one observed in magneto-optical materials, but in which optical pumping with given handedness replaces the magnetic bias.

## Results

**Photoinduced nonreciprocal circular dichroism.** The scheme illustrating the concept of photoinduced nonreciprocity is shown in Fig. 1a. A TMD monolayer is pumped by a strong CP laser radiation, which leads to the selective formation of exciton gas at one of the valleys (Fig. 1b). Provided the valley polarization of excitons is at least partially preserved, it will give rise to an asymmetric response of the TMD monolayer to weak probe signals of opposite handedness, due to the fact that they selectively interact with one of the two valleys[35–38]. As schematically depicted in Fig. 1a, this effect leads to a nonreciprocal dichroic response, i.e., probe signals of opposite handedness are absorbed and reflected differently from the optically pumped TMD monolayer. Since the handedness of CP light is locked to the propagation direction, and similar locking of transverse angular momentum to propagation direction exists for evanescent electromagnetic fields[57–59], this opens the opportunity for designing optical elements with inherently nonreciprocal response induced by the CP pump field.

As an example, we consider the optical response of a WS$_2$ monolayer under CP pump, expecting to observe nonreciprocal circular dichroism at large pump intensities. As schematically shown in Fig. 1a, b, a pump with right-handed circular polarization ($\sigma^-$) illuminating the sample from the right side (positive $k_z$ direction) has a clockwise (CW) projected helicity on the plane of the 2D material, which leads to the formation of excitons at the K′ valley in WS$_2$ (Fig. 1c). In the ideal case of no valley scattering, this increased exciton density at one of the valleys affects reflectivity of a probe signal of the same $\sigma^-$ helicity incident from the right side ($k = +|k_z|$) of the sample, because the probe beam has the same (CW) projected helicity as the pump (Fig. 1a). On the other hand, the reflectivity of the $\sigma^-$ probe field incident from the opposite (left) side ($k = -|k_z|$), which has the counterclockwise (CCW) projected helicity opposite to that of the pump and thus interacts with the opposite K-valley (Fig. 1d), remains unaffected (Fig. 1b). As a consequence, the two probe signals illuminating from the opposite sides will experience different absorption, reflection, and transmission. This nonreciprocal response (e.g., $r(+k_z) \neq r(-k_z)$) can be used to realize an all-optical magnet-free isolator, as shown in the following.

**Theoretical model.** In order to quantitatively describe the dichroic nonreciprocal response in optically pumped TMDs, we introduce the pump intensity dependent surface conductivity tensor (in the linearly polarized basis)

$$\hat{\sigma} = \sigma_N(I) + \hat{\sigma}_K(I_{CW}) + \hat{\sigma}_{K'}(I_{CCW}), \tag{1}$$

where $\sigma_N$ describes the valley-independent optical response, $\hat{\sigma}_K$, $\hat{\sigma}_{K'}$ correspond to the valley-dependent response due to excitations at K and K′ valley, respectively, $I_{CW}$, $I_{CCW}$ are the intensities of the CP pump fields, which make CW and CCW projections of the electric field onto the TMD plane, respectively, and $I = I_{CW} + I_{CCW}$ is the total pump field. In addition to conventional non-valley polarized optical processes, the first term in Eq. (1), $\sigma_N(I)$, also accounts for valley "depolarization" due to various intervalley scattering processes. It is worth highlighting that here the notations of CW and CCW specify the handedness of the electric field rotation in the TMD plane, irrelevant of the propagation direction, and that LCP ($\sigma^+$) and RCP ($\sigma^-$) polarization/handedness of optical waves are therefore not in one-to-one correspondence

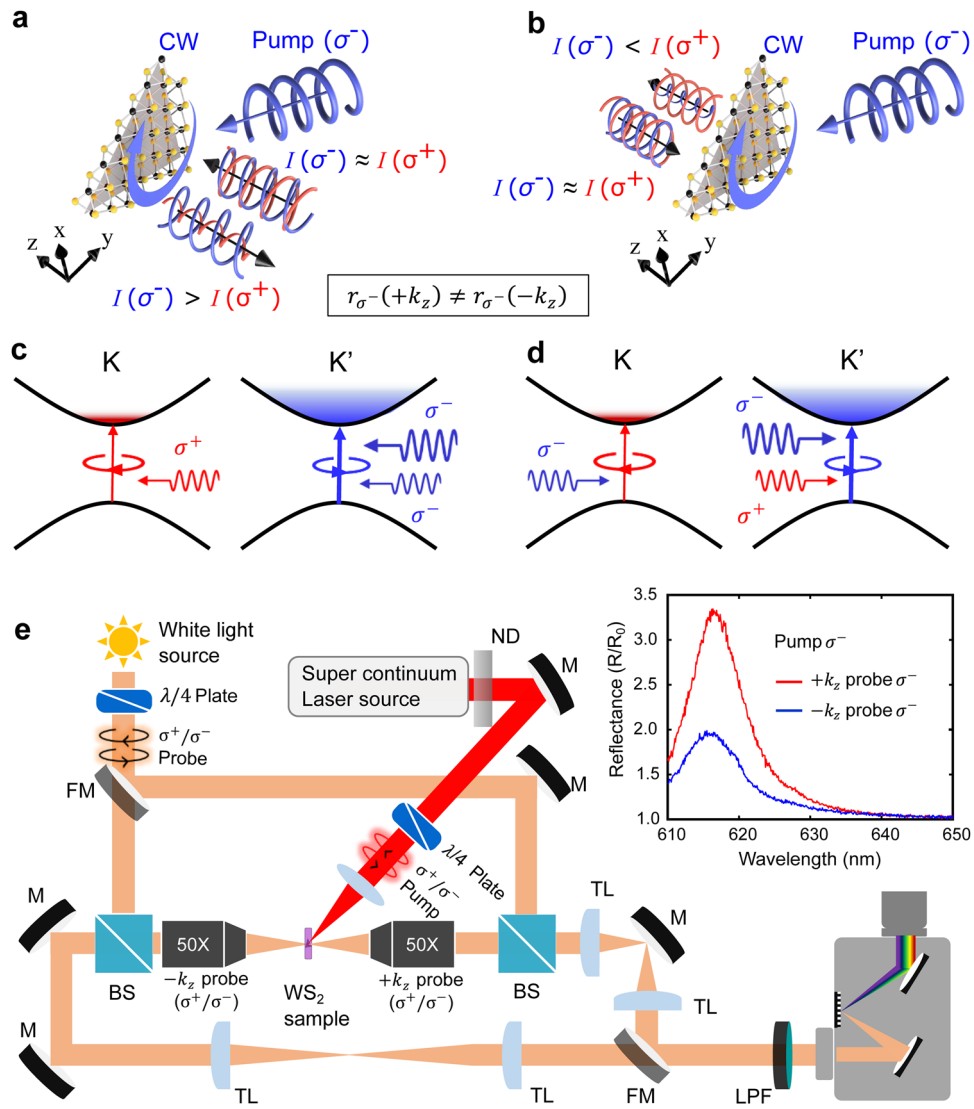

**Fig. 1 Photoinduced nonreciprocity in a TMD monolayer. a, b** Schematic illustration of nonreciprocal reflection due to valley-selective response induced by a CP pump. A clockwise (CW) pump is shown as an example. **c, d** Increased exciton population density at K' valley due to $\sigma^-$ pump is probed by beams of different helicities incident from the opposite directions. **e** Schematic of the experimental set up to probe reflection of a TMD monolayer under CP pump (as in **a** and **b**) in wavevector inversion geometry. Here, M mirror, FM flip mirror, L Lens, TL tube lens, BS beam splitter, ND variable neutral density filter. **f** Nonreciprocal reflectance response of WS₂ monolayer for $\sigma^-$ probe beams propagating in $+k_z$ (red curve) and $-k_z$ (blue curve) directions for intense $\sigma^-$ pump incident from one side ($+k_z$ direction).

with CW and CCW. The valley-polarized response is uniquely described by the projected helicity (CW/CCW).

The valley-dependent terms in (1) have the following form, which accounts for their chiral response:

$$\hat{\sigma}_K = \frac{1}{2}\begin{pmatrix} \sigma_K & i\sigma_K \\ -i\sigma_K & \sigma_K \end{pmatrix}, \hat{\sigma}_{K'} = \frac{1}{2}\begin{pmatrix} \sigma_{K'} & -i\sigma_{K'} \\ i\sigma_{K'} & \sigma_{K'} \end{pmatrix}, \quad (2)$$

where $\sigma_K = \sigma_K(I_{CW})$ and $\sigma_{K'} = \sigma_{K'}(I_{CCW})$ are surface conductivities for the two valleys in the CP basis. The form of Eq. (2) follows directly from the fact that the response of each valley is selective with respect to the handedness of the optical field, and therefore it is described by the matrices $\hat{\sigma}_K^{CP} = [\sigma_K, 0; 0, 0]$ and $\hat{\sigma}_{K'}^{CP} = [0, 0; 0, \sigma_{K'}]$ in the CP basis.

In the case of no optical pump, the two valleys yield the same response $\sigma_K(I_{CW} = 0) \equiv \sigma_{K'}(I_{CCW} = 0)$, so that $\hat{\sigma}_K + \hat{\sigma}_{K'} = [\sigma_K, 0; 0, \sigma_{K'}]$ and the TMD shows no asymmetry in the response with respect to $\sigma^+$ and $\sigma^-$ probe signals. However, dichroism arises as the pump intensity of a particular handedness is

increased, and the response enters the valley-polarized regime such that $\sigma_K(I_{CW}) - \sigma_{K'}(I_{CCW}) \neq 0$, yielding an effective response of the form

$$\hat{\sigma}_{TMD} = \begin{pmatrix} \sigma_N + \frac{1}{2}(\sigma_K + \sigma_{K'}) & \frac{i}{2}(\sigma_K - \sigma_{K'}) \\ -\frac{i}{2}(\sigma_K - \sigma_{K'}) & \sigma_N + \frac{1}{2}(\sigma_K + \sigma_{K'}) \end{pmatrix} = \begin{pmatrix} \sigma_{xx} & i\sigma_{xy} \\ -i\sigma_{xy} & \sigma_{xx} \end{pmatrix}. \quad (3)$$

This response is equivalent to the one of a 2D electron gas in the presence of a dc external magnetic bias[60,61], showing how the CP optical pump can effectively break time-reversal symmetry in TMDs. Indeed, the possibility to use a CP pump as an effective magnetic field bias has been considered before in the context of so called Floquet systems and Floquet topological insulators[28,62,63]. More recently, the CP pump field was used to demonstrate photoinduced quantum Hall effect in graphene[64]. Here we report the realization of nonreciprocal optical response by exploiting the valley degree of freedom in 2D TMDs.

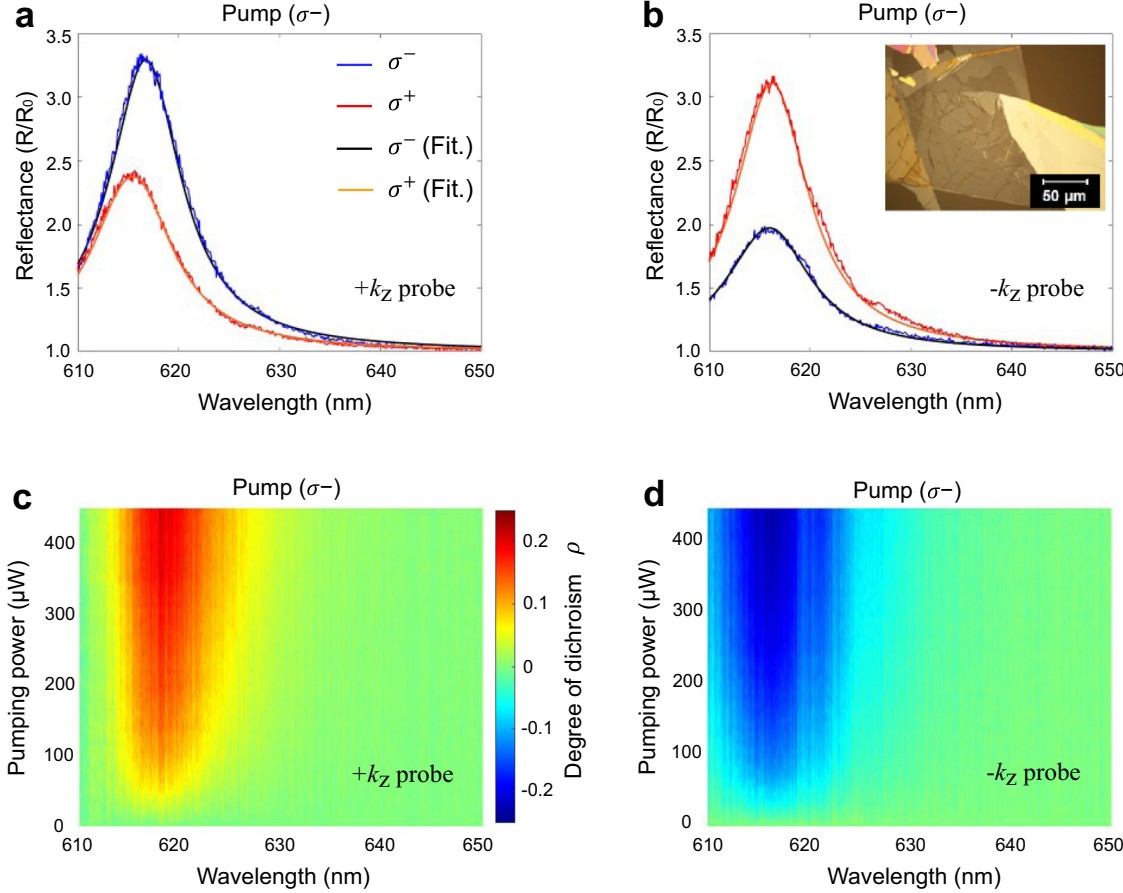

**Fig. 2 Experimental demonstration of photoinduced nonreciprocal circular dichroism in WS₂. a, b** show cases of $\sigma^+$ and $\sigma^-$ probes spectral reflectivity, under wavevector inversion geometry (for $k = k_z$ and $k = -k_z$ cases), for the $\sigma^-$ pump incidence. Solid lines show the result of fitting by a modified Fresnel equation with surface conductivity described by a Lorentzian model (see Methods for details). Inset shows the optical microscope image of an encapsulated WS₂ monolayer on a glass substrate. **c, d** show pump power dependent dichroic response as seen in reflectivity from the sample (see Eq. 4) for probes coming from opposite directions. Here 600 nm $\sigma^-$ pump is incident as in positive $k_z$ direction (as shown in Fig. 1). The results for the $\sigma^+$ pump can be found in the Supplementary Note 2.

We note that a different form of reciprocal dichroic response can be found in planar systems and can be attributed to nanoscale patterning and substrate effects[65–69], but it cannot be used for nonreciprocal device applications.

**Experimental results**. In order to experimentally verify the photoinduced dichroic response, we have performed measurements on a WS₂ monolayer encapsulated between two thin hBN layers of 7 nm each and transferred onto a glass substrate. The image of the sample is shown in Fig. 2b inset (details of the sample preparation can be found in Methods). To ensure that the reported results do not originate from asymmetry of the geometry, e.g., the substrate induced bianisotropy, the sample was embedded into a symmetric dielectric environment. First, it was encapsulated between two hBN layers of the same thickness, and, second, it was coated with a thick PMMA (A11) layer, which has the same refractive index as the BK7 glass substrate in the wavelength range of interest.

The sample was pumped by 190 ps CP pulses from a supercontinuum source (NKT SuperK SELECT) with 12 ns pulse period at the wavelength on the blue-side and as close as possible to the exciton resonance (616 nm) to promote formation of excitons. The intensity of the CP pump was gradually increased, and the sample reflectivity was probed with low-intensity CP beams from a halogen light source from two opposite directions.

We collected the reflected signal in back focal plane imaging configuration and analyzed with CCD spectrometer as shown in Fig. 1e (see Methods section for details).

The possibility of wavevector inversion in our setup allows to experimentally test the nonreciprocity due to the photoinduced dichroic response. First, the comparison of Fig. 2a, b, which show reflectivities collected for the two opposite propagations of probe signals at the pump power of 450 μW, clearly demonstrates nonreciprocal reflection of the probes of the same handedness coming from the opposite sides. Second, the more detailed picture of this phenomenon is provided Fig. 2c, d. The latter show the pump power dependent dichroic response obtained for the probe beams of two opposite helicities incident in the $+k_z$ or $-k_z$ direction and for the $\sigma^-$ pump incident in $+k_z$ direction. The degree of dichroic response for each direction is calculated from the following expression

$$\rho(\pm k_z) = \frac{R^{\sigma^-}(\pm k_z) - R^{\sigma^+}(\pm k_z)}{R^{\sigma^-}(\pm k_z) + R^{\sigma^+}(\pm k_z)} \quad (4)$$

Figure 2c, d show that, as the pump intensity increases, the dichroic response of probe signals with opposite projected helicities become increasingly different. This pump induced dichroic responses exhibit contrary responses for the probes of the same handedness incident from the opposite directions, thus confirming our hypothesis about nonreciprocity of the optical

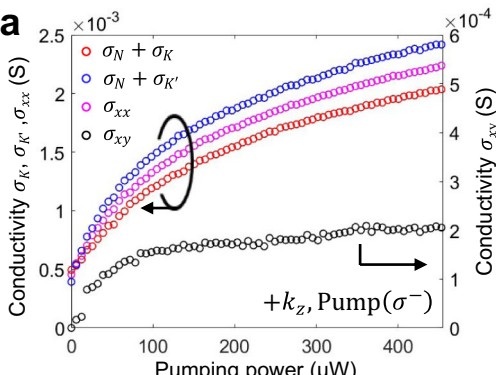
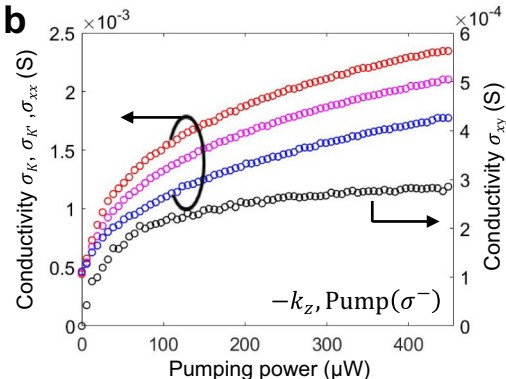

**Fig. 3 Measured dichroic surface conductivity of WS₂ under CP optical pump. a, b** Surface conductivities obtained from experimental data by the fitting reflectance for both $+k_z$ and $-k_z$ directions of the CP probes. Diagonal element of the surface conductivity plotted alongside with the off-diagonal element. The dots in **a, b** show values extracted from experimentally measured reflectivities. (Supplementary Note 1).

dichroism. As seen from Fig. 2a–d, the dichrism shows the largest increase at the frequency of the exciton resonance (plotted separately in Supplementary Figs. 3 and 4), which farther proves that the mechanism responsible for the nonreciprocity is associated with the difference in exciton densities at the two valleys. To explain the observed photoinduced nonreciprocal dichroic behavior, we developed an analytical model based on the Fresnel equations modified by the introduction of a TMD monolayer, whose optical response is described by a surface conductivity with Lorentzian dispersion[53,70,71] (see Supplementary Note 1 for details). In addition, we incorporated the valley-polarization related effects into the Lorenz model by accounting for an increase in the exciton density.

Some asymmetries between forward and backward data in Fig. 2 (and Fig. 3 below) is attributed to a slightly different position of the 5 μm diameter probe beam on the sample, which gives rise to a relative shift with respect to position of the pump beam (gaussian beam with 25 μm diameter) and thus slightly different local exciton valley population. In addition, a regular nonuniformity of valley polarization due to an inhomogeneity of the monolayer can play some role here[72]. Indeed, as can be seen in Fig. 2a, the exciton peaks are shifted by ~1 nm with respect to one another, which can be attributed to local strain in the TMD monolayer. Despite these nonuniformities, the nonreciprocity is clearly visible and is highly pronounced.

The proposed model was used to fit the experimental data, enabling the retrieval of the surface conductivity tensor, with results shown in Fig. 3. Since the conductivity for each of the projected helicities is $\sigma_K = (\sigma_{xx} - \sigma_N) + \sigma_{xy}$ and $\sigma_{K'} = (\sigma_{xx} - \sigma_N) - \sigma_{xy}$, our extracted photoinduced off-diagonal (dichroic) component of the surface conductivity reaches the value $\sigma_{xy} = (\sigma_K - \sigma_{K'})/2 \approx 0.17\,\sigma_{xx}$ at the peak reflectivity. This value of dichroic response is impressive, considering that the measurements are performed at room temperature, in which case the intervalley scattering is expected to play a detrimental role. In addition, the 190 ps pulse duration of the pump signal with 12 ns repetition rate limits the overall valley-polarized response. The fact that the dichroism does not vanish over such long integration times (compared to pump duration) indicates the presence of long-lived valley-polarized excitations in the system.

Indeed, while the excitons in TMDs are known to have rather shorter lifetimes of less than 2 ps, recent time-resolved pump-probe experimental studies have suggested that the lifetime of photoexcited free carriers can be as long as few ns[73–76] at room temperature, and may exceed values of 10 ns at cryogenic temperatures[77] and reach 130 ns if samples are electrically gated[78]. Such long relaxation times was attributed to the presence

of valley polarized resident carriers[75–79]. Thus, we suggest that a mechanism behind the observed circular dichroism is associated with the delayed relaxation of the photoexcited valley-polarized free carriers into exciton states with partial preservation of the valley-polarization. In the proposed scenario, the valley-preserving relaxation of free-carriers leads to a larger density of excitons at one of the valleys, giving rise to stronger absorption of light of a particular handedness (Supplementary Note 2, section 2.8). Considering that we use a pulsed excitation, we expect that even stronger nonreciprocal dichroic response may be found for continuous wave pump, which can make our proposed approach to nonreciprocity even more viable for practical applications.

**Proposed device scheme of an all-optical isolator.** To demonstrate that the photoinduced circular dichroism phenomenon can yield nonreciprocal operation, we propose a practical design of a magnet-free optical isolator relying on this effect. The proposed device is based on a silicon nitride ring resonator critically coupled to a waveguide; this scheme was recently employed for high speed modulation of light with 2D materials, graphene[80], and tungsten disulfide WS₂[81]. The functionality of the device is illustrated in Fig. 4a, b and it is based on spin-Hall effects[82] of light associated with the nonvanishing transverse angular momentum of the evanescent optical field of guided waves[57–59]. In particular, the mode guided in the forward direction by a SiN waveguide is evanescent in the cladding, and it is characterized by CW (CCW) elliptically polarized nearfields on the right (left), as schematically shown in the inset to Fig. 4a. If the propagation direction of the guided wave is reversed, as in the inset to Fig. 4b, the handedness of the evanescent field accordingly reverses. Therefore, by placing a dichroic TMD monolayer asymmetrically with respect to the waveguide (only on the one side, as in Fig. 4a, b insets/zoom-ins) we expect different absorption rates for oppositely propagating guided waves. Indeed, due to the dichroic response in optically pumped TMD, the different nearfield overlap of the guided wave with the surface conductivity of TMD must yield a different absorption rate for forward and backward guided modes. Such nonreciprocal absorption can be estimated using electromagnetic perturbation theory[83,84].

Taking the electric field $\mathbf{E}_0$ of the guided mode without the TMD monolayer as the unperturbed solution, and treating the monolayer as a perturbation, the attenuation rate due to the absorption in the TMD can be estimated to first-order to be

$$\text{Im}(\beta) = \frac{\beta_0}{\omega W_0} \int_{\text{TMD}} dS[\mathbf{E}_0^* \text{Re}(\hat{\sigma}_{\text{TMD}})\mathbf{E}_0], \qquad (5)$$

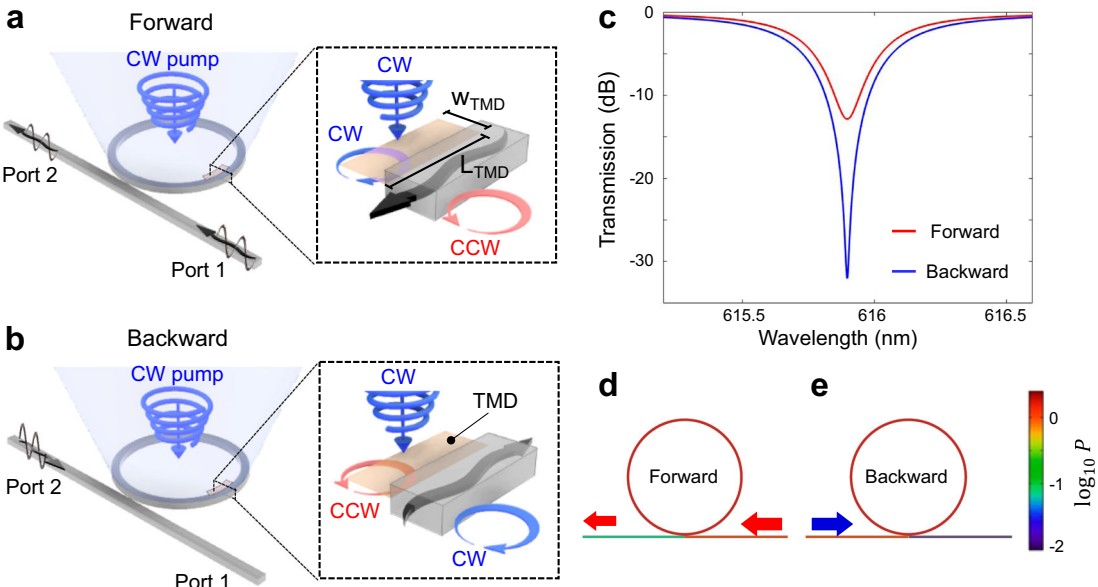

**Fig. 4 All-optical isolator device design and its operation principle. a, b** SiN ring resonator loaded asymmetrically by a transition metal dichalcogenide (TMD) monolayer (on the inner side of the ring only to maximize asymmetric loss), which explains nonreciprocal transmission due to different absorption of clockwise (CW) and counter-clockwise (CCW) modes in the ring resonator. **c** Nonreciprocal transmission through the waveguide with forward and backward transmission shown by blue and red lines, respectively. The ring radius $R = 15$ μm, therefore the trip distance $L = 2\pi R$, t = 0.816, and $a_+ = 0.824$, $a_- = 0.888$, which correspond to the experimentally extracted surface conductivity tensor $\sigma_{xx} = 1.10 \times 10^{-3}$ and $\sigma_{xy} = 1.92 \times 10^{-4}$ with parameters $\alpha = 4.69 \times 10^4$ 1/m, $\delta = 1.13 \times 10^4$ 1/m, $L_{TMD} = 3.3$ μm, the width of the TMD monolayer is $w = 50$ nm. **d, e** Power density in the all-optical isolator for forward (**d**) and backward (**e**) propagating waves, respectively, as calculated from coupled mode theory (CMT).

where $\beta_0$ is the unperturbed wavenumber of the guided wave, $S$ is the surface area, and $W_0 = 2 \int_V dV[|E_0(r)|^2 \epsilon_0 \epsilon_r(r)]$ is the energy density of the unperturbed guided wave, with the integration performed over the mode volume. In this discussion, we neglect the small material loss in the unperturbed waveguide. Decomposing the evanescent field in the TMD region into the CW and CCW helicity components $\mathbf{E}_0 = \mathbf{E}_{0CW} + \mathbf{E}_{0CCW}$, we obtain $Im(\beta) = Im(\beta_0) + Im(\beta_K) + Im(\beta_{K'})$ where $Im(\beta_0) \equiv \alpha = \frac{\beta_0}{\omega W_0} \int_{TMD} dS[|E_0|^2 \sigma_{xx}]$ and $Im(\beta_{K/K'}) = \frac{\beta_0}{\omega W_0} \int_{TMD} dS[|E_{0CW/CCW}|^2 \sigma_{K/K'}]$ are valley independent and valley polarized contributions to the attenuation caused by the absorption in the TMD. Since the pumped TMD monolayer has $\sigma_K \neq \sigma_{K'}$ and due to the evanescent field of the guided mode is chiral, we obtain a nonzero differential attenuation for forward and backward waves: $\delta \equiv Im(\beta(k>0) - \beta(k<0))/2 = \frac{\beta_0}{2\omega W_0} \int_{TMD} dS[(|E_{0CW}|^2 - |E_{0CCW}|^2) (\sigma_K - \sigma_{K'})]$. By applying this analysis to the modal solution for the SiN waveguide obtained with COMSOL Multiphysics, we retrieved $Im(\beta_{backford}) = 5.82 \times 10^4$ 1/m and $Im(\beta_{forward}) = 3.56 \times 10^4$ 1/m. The magnitude of such nonreciprocal attenuation, however, is not large enough to yield sufficient isolation for reasonable propagation distances. We therefore employ a resonant scheme to enhance the nonreciprocal differential absorption.

The optical isolator layout is shown in Fig. 4a, b and it consists of a SiN waveguide evanescently coupled to a SiN ring resonator. We place the circularly pumped TMD monolayer on the inner side of the ring resonator only, which ensures that the modes propagating in opposite directions have different attenuation rates. Indeed, similar to the case of the waveguide, the evanescent component of the electric field of the mode in the ring resonator carries angular momentum of opposite handedness on the inner and outer sides of the ring. The handedness of the evanescent fields flips when the propagation direction in the ring resonator reverses from CW propagating mode to CCW propagating mode, which again gives rise to a difference in absorption for the two modes.

Such different absorption rates for modes propagating in opposite directions enable a selective critical coupling between the ring resonator and the waveguide[85,86] only for one propagation, yielding a strong nonreciprocal response. From here on we will use subscripts +/– to indicate CCW and CW propagation directions in the ring resonator to avoid any confusion with the notations of the projected handedness of the electric field of the modes on the TMD.

According to CMT[87], the critical coupling condition for the mode propagating backward, i.e., from Port 2 to Port 1 in the SiN waveguide (Fig. 4b), and therefore coupling to the CCW (+) mode of the ring resonator, is $t = a_+$. Here $t$ is the self-coupling of the waveguide and $a_+ = \exp[-(\alpha + \delta)L_{TMD}]$ is the round-trip loss coefficient in the ring resonator for the CCW(+) mode, and $L_{TMD}$ is the length of coverage of the ring resonator by the TMD monolayer. This condition cannot be satisfied simultaneously for the waveguide mode propagating in the forward direction, i.e., from Port 1 to Port 2, thus yielding the nonreciprocal transmission. In the latter case the critical coupling condition is $t = a_-$, where $a_- = \exp[-(\alpha - \delta)L_{TMD}]$, since the guided mode now couples to the CW (–) mode in the ring resonator which has different round-trip loss coefficient $a_- \neq a_+$.

To confirm the functionality of the proposed device, we performed CMT[87] modeling with the parameters obtained from perturbation theory using the field profiles $\mathbf{E}_0$ for unperturbed SiN waveguide calculated in COMSOL. For the conductivity parameters retrieved from our experimental data, we found that an isolation of 20 dB with ~7% of forward transmission can be readily achieved. The corresponding results showing (i) forward transmission $S_{12}$ (from Port 1 to Port 2) and (ii) backward transmission $S_{21}$ (from Port 2 to Port 1) are plotted in Fig. 4c and clearly reveal a strong nonreciprocal response. The corresponding field profile found with the use of CMT for forward and backward incidence and illustrating nonreciprocal operation at the frequency of near-critical coupling in the backward direction are shown in Fig. 4d.

We note that higher values of isolation are possible at the expense of lower forward transmission. The main limiting factor for even stronger isolation is the intervalley scattering, which gives rise to the non-dichroic loss in the device and suppression of forward transmission. Intervalley scattering can potentially be significantly reduced by lowering the temperature, which would yield lower non-valley selective component of the surface conductivity $\sigma_N$. However, we believe that, even in the demonstrated example operated at room temperature, the proposed approach to nonreciprocity can already be more practical than the use of magneto-optical materials in many applications.

## Discussion

To summarize, we experimentally demonstrated the emergence of a nonreciprocal dichroic optical response in WS$_2$ monolayer biased by CP pump field. The dichroic response is explained as the result of interaction of light with valley polarized excitons, whose increased population at one of the valleys for higher CP pump intensities leads to disparate scattering of the weak probe fields of opposite handedness.

The analogy of the observed dichroic response with the one of 2D electron systems in an external magnetic field suggests the possible use of optically pumped TMDs to produce magnet-free nonreciprocity. A device based on locking of transverse angular momentum of the evanescent field with the propagation direction of the guided waves was proposed. Asymmetric absorption rates due to chiral light–matter interactions with a dichroic TMD monolayer placed on SiN ring resonator was shown to be a viable mechanism to achieve unidirectional critical coupling, thus producing optical isolation.

Unlike the magnets in conventional bulky isolators, the pumping laser can be fully integrated on chip in our proposed device operation scheme (Supplementary Note 2, section 2.7), which paves the way toward fully integrated nonreciprocal photonic systems. Indeed, the effect reported here appears at rather moderate pump fluences. In our experiment we have 450 μm maximal pump beam focused on 25 μm × 25 μm area, which, for the integrated setup in Fig. 4 (with TMD of size 3.3 μm by 50 nm) implies the reduction of power by ~3000 times, which corresponds to the power of 0.2 μW. This power can be farther reduced by using an integrated scheme (Supplementary Note 2, section 2.7), where the pump field enhanced in the ring resonator itself. Such fully integrated setup offers tunability by switching on/off the pump fields, effectively yielding an all-optical control of nonreciprocity and thus it has clear advantages over current static magnetic nonreciprocal devices.

We believe that the recent progress in integration of 2D materials with existing photonic materials and devices[81] will facilitate introduction of the proposed magnet-free approach to all-optical nonreciprocal into practical systems. Therefore, this approach envisions a new generation of all-optical isolators and circulators integrated into on-chip photonic systems. Furthermore, the possibility to control the direction of optical isolation, by simply switching the handedness of the pump, makes these nonreciprocal devices switchable on the fly—property hardly achievable in conventional magnetic devices—thus enabling applications in classical and quantum photonic frameworks.

## Methods

**Sample fabrication**. A monolayer of WS$_2$ TMD material and two thin hBN layers each of ~7 nm thickness were exfoliated onto a thick PDMS stamps using standard tape technique and transferred to 120 μm thick silica substrate by home build transfer stage one after another. Both the hBNs and the WS$_2$ monolayer were annealed at 350 °C for 3 h in N$_2$ atmosphere prior to the next layer transfer to remove the PDMS residue (from the transfer process). Finally, the sandwiched WS$_2$ and hBN stack on the glass substrate was coated with 1.7-μm-thick 495 PMMA (A11) polymer film followed by annealing at 180 °C for 1 min.

**Experimental set up**. High-intensity supercontinuum light-source SuperK Extreme with connected SuperK SELECT tunable high-resolution bandpass filter generated light beam with 2 nm bandwidth and the tunable wavelength in the range 0.4–1.0 μm. The sample was pumped with supercontinuum pulsed laser of 190 ps pulse width and 12 ns pulse period and wavelength of light was set to 600 nm close to exciton resonance (616 nm) at room temperature. The polarization of the excitation beam was set to circular polarization by using a combination of linear polarizer and quarter wave plate. A plano-convex lens with 5 cm focal length was used to pump the monolayer WS$_2$ with 25 μm spot size at 30° angle of incidence. A variable neutral density filter was used to vary the incident pump power on the sample. A low intensity white light beam from halogen light source was used as a probe beam with circular polarization set up by a set of linear polarizer and quarter wave plate. The CP white light probe beam was directed towards two inverted wavevector directions using a flip mirror as shown in Fig. 1d. Two 50X microscopic objectives (BoliOptics and Olympus) with 5 μm spot size at the focus were used to probe the sample reflection spectrum from two opposite directions to demonstrate nonreciprocity. A complete setup was bult as 4f configuration system to image the back focal plane of both the objectives onto the entrance slit of CCD detector coupled spectrometer as shown in Fig. 1d. A 610 nm long pass filter was used to cut-off the pump beam prior to the spectrometer entrance slit.

## Data availability

The data that support the findings of this study are available from the corresponding author upon reasonable request.

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

## Acknowledgements
The work was supported by the National Science Foundation with grants No. DMR-1809915, NSF ECCS-1906096, by the Defense Advanced Research Project Agency Nascent Program, by the Air Force Office of Scientific Research MURI program, and by the Simons Foundation.

## Author contributions
All authors contributed extensively to the work presented in this paper. A.B.K. conceived the idea. S.G., F.K., S.K., and A.V. prepared samples and performed the experiments. A.B.K., Y.K., and K.C. developed the models and fitted experimental data. A.B.K., A.A., and V.M. supervised work. All coauthors contributed to writing the manuscript. S.G. and Y.K. contributed equally to this work.

## Competing interests
The authors declare no competing interests.
