## [Peer Review File · Nature Communications]

Reviewers' comments:

Reviewer #1 (Remarks to the Author):

The manuscript by Yuma Kawaguchi et al is interesting and based on convincing motivation. However I have problems with finding sufficiently new results important for the field of physics. I have doubts whether the overall novelty of the presented research justifies publication in Nature Communications. The authors propose utilization of semiconductor transition metal dichalcogenides in design of devices with optically induced nonreciprocity. The idea is based on the phenomenon of optically induced circular dichroism. The authors present experimental results proving appearance of such dichroism in WS₂ samples. However the observation of such dichroism is not really new. Actually circular dichroism was studied for many years by means of pump-probe experiments including measurements of time resolved Faraday and Kerr rotation. The investigated systems were different including DMS-s layers, DMS quantum wells, doped semiconductor quantum wells and different semiconductor materials including TMDs. For certain examples the relaxation time was shown to be quite long. Therefore, in my opinion, the experimental data presented in the manuscript is neither new nor represents real experimental state of the art.

The proposed simple theoretical model is phenomenological and gives no real insight into physics of exciton-exciton or exciton-carrier interactions. The microscopic mechanism of the long-lasting circular dichroism remains unknown. In particular it is not clear whether the samples under examination are doped (p or n type) or the leading mechanism is related to dark exciton populations. The claim that the microscopic picture is illustrated in Fig. 1b is exaggerated. In my opinion any conclusions would require more careful analysis taking into account dynamics of different processes (relaxations and lifetimes of different excitonic complexes). Therefore this part of the manuscript seems to be speculative and nebulous.

The main novelty of the paper resides in proposal of the design of optical isolator device. Unfortunately the proposal is not supported by any experimental efforts which strongly limits its value. In particular the time integrated results obtained in pump-probe experiment should be used with caution. In my opinion they cannot be used directly as a starting point for the device analysis. The operation of the real device might be strongly affected by temporal variation of optical properties. The correct analysis should be based on experiments with continuous wave excitation.

Additionally to my general objections I have a few minor remarks:

The schematic drawings are not readable - the circular symbols representing circularly polarized light dominate everything making other elements less visible. Particularly the device design in fig 4 is not clear. The abbreviations CW and CCW might be somehow misleading as CW laser usually means "continuous wave" operation, not polarization.

Taking into account all weakness I believe that the work in its present form should not be published in Nature Communications.

Reviewer #2 (Remarks to the Author):

Yuma Kawaguchi and co-authors experimentally demonstrate photoinduced nonreciprocity in transition metal dichalcogenides (TMDs) pumped by circularly polarized light. The authors claim that the nonreciprocity stems from the valley-selective exciton-exciton interactions that result in the nonlinear circular dichroism. Because the nonreciprocity is achieved without the involvement of an external magnetic field, it may be useful for future all-optical integrated photonics. As an example, they propose an optical isolator based on the optically biased TMD monolayer. The results are convincing, timely, and important to the field of nonreciprocal optics and 2D materials. Besides, the manuscript is very well-written, and the graphs are clear of understanding. In terms of the quality of the text, the manuscript fulfils the standards of Nature Communications. Therefore, I would like to recommend publication of this work in Nature Communications. I only have some minor suggestions on the manuscript.

1. In the introduction, the authors state that the drawback of the conventional nonreciprocal optical devices

based on magneto-optical materials is the bulky external magnetic bias. However, their approach proposed in this work relies on an external laser source for optical pumping. Is it also bulky?

2. In the second paragraph of the introduction, the authors state that "However, this form of self-bias nonreciprocity comes with some drawbacks that hinder its widespread applicability." Can the authors make it clear that what the drawbacks are?

3. In the experiments, the authors pump the TMDs with an intense circularly polarized laser radiation and illuminate the TMDs with a circularly polarized probe signal. They found that "the probe signals illuminating from opposite angles with opposite helicity, time-reversed versions of each other, will experience different reflectivity." Can the probe signals be replaced with linearly polarized light? What is the reflectivity expected to see? I think this is important because circular dichroism can also arise from the structure chirality.

4. Below Eq. (2), it says "where $\sigma_K = \sigma_K(ICW)$ and $\sigma_{K'} = \sigma_{K'}(ICW)$ are surface conductivities for the two valleys in the circularly polarized (CP) basis. The form of Eqs. (2) follows directly from the fact that the response of each valley is selective with respect to the handedness of the optical field, and therefore it is described by the matrices $\hat{\sigma}_{KCPB} = [\sigma_K, 0; 0, 0]$ and $\hat{\sigma}_{K'CPB} = [0, 0; 0, \sigma_{K'}]$ in the circularly polarized basis (CPB)." The first "circularly polarized (CP) basis" should be "linearly polarized basis"? The second "circularly polarized basis (CPB)" should be "circularly polarized basis (CPB)".

5. On page 7, "as in Fig. 3a,b insets/zoom-ins" should be "as in Fig. 4a,b insets/zoom-ins."

6. In Fig. 4, can the authors show the field patterns in the waveguide and ring resonator to better understand the operation of the optical isolator?

Reviewer #3 (Remarks to the Author):

This manuscript reports on the use of photo-excited 2D materials (WS₂) as the basis of a magnet-free non-reciprocal devices for photonics. After an introduction to remind the state-of-the-art, the manuscript consists of three main sections: (i) description and theoretical modeling of the proposed effect ; (ii) an experiment to demonstrate the existence of the effect ; and (iii) the design of a nonreciprocal device, an optical insulator (a study that remains purely theoretical at this stage).

The main idea of the paper is that a single layer of transition metal dichalcogenide, excited by an intense and circularly polarized pump at the excitonic resonance wavelength, will exhibit a polarization-dependent reflectivity to a probe beam (as sketched in Fig. 1a). This claim sounds convincing, and an experiment is proposed to support it (Figure 2). My concern (and main criticism) is that although this figure is entitled "Experimental demonstration of photo-induced nonreciprocal circular dichroism", I do not think that the claim of a nonreciprocal behavior is demonstrated at all in this experiment. From what is reported in the manuscript, I understand that the authors demonstrate photo-induced circular dichroism (a very interesting and important result by itself) but NOT photo-induced nonreciprocal circular dichroism. To demonstrate nonreciprocity, some kind of wavevector inversion should be involved, and the experimental setup (shown in the supplementary information) do not allow that. Contrary to what is sketched in Fig. 1a, the actual experiment does not involve any change of the parallel component of the probing beam's wavevector. The probe beam impinge the sample in normal incidence and the reflected light is collected through the same objective lens. I fail to see how such an experiment could demonstrate anything about reciprocity.

If I understand correctly, the point made by the authors is as the sample is a 2D material, then circular dichroism must be nonreciprocal, as stated page 3 of the manuscript: "Is worth noting here that the circular dichroic response of any planar 2D systems, including 2D materials, is necessarily nonreciprocal, since reciprocal circular dichroism, known as optical activity, requires nonlocal bianisotropic response, which is possible only in structures with finite thickness". This claim is supported by two bibliographical references. I had a look at these papers:

- reference 57 does not evoke reciprocity at all;
- reference 56 does evoke reciprocity, but does not provide direct evidence to the assumption made by the authors.

As a consequence, I do not think this assumption is well-supported, and a more convincing demonstration

must be provided.

Furthermore, there might be another problem. Even if the above assumption is correct, the actual sample studied in the manuscript is not 2D: it is a 2D layer sitting on a substrate. The potential role played by the substrate is completely eluded in the whole manuscript. Researchers working in the field of chiral plasmonics have shown, when working with planar metallic nanostructures, the critical role played by the substrate. This question actually led to a strong debate in the community - see, e.g. Kuwata-Gonokami, M. et al. Giant Optical Activity in Quasi-Two-Dimensional Planar Nanostructures. *Phys. Rev. Lett.* 95, 735 (2005) ; Collins, J. T. et al. Chirality and Chiroptical Effects in Metal Nanostructures: Fundamentals and Current Trends. *Advanced Optical Materials* 5, 1700182 (2017) ; Drezet, A. & Genet, C. Reciprocity and optical chirality. *arXiv.org physics.optics* (2017). It is now generally admitted that "planar chirality" is associated to the symmetry-breaking brought by the substrate. I cannot say if such effects are involved here, but the question must be discussed in the manuscript. As a relevant example, the authors write on page 4 of the manuscript: "It is worth highlighting that here the notations of CW and CCW specify the handedness of the electric field rotation in the TMD plane, irrelevant of the propagation direction". Does this assumption still hold if there is a substrate? I would say that the handedness is not the same looking from the air side or from the substrate side. It is worth noting that the directions from which the sample is pumped and probed are not specified.

To conclude, this manuscript reports on a very interesting experiment on photo-induced circular dichroism. This result most certainly deserves publication and it might be appropriate for *Nature Communications*. However, in my opinion publication at this stage would be premature. Either further experiments are required to demonstrate the nonreciprocal behavior, or the proposed interpretation is not clearly exposed: in any case a substantial rewriting appears necessary to clarify the two issues I mentioned above.

Other comments/questions:

1. The schematic of the experimental set-up should be moved from the supplementary information to the main text. It is of critical importance to understand the experiment.
2. On page 3, the authors write: "reciprocal circular dichroism, known as optical activity"; I am not sure to understand correctly. Optical activity refers to any chiroptical effect involving a different response to right- and left-circularly polarized light (see L.D. Barron, *Molecular light scattering and optical activity*, Cambridge Univ. Press, page 1). Hence, optical activity includes circular dichroism (as well as optical rotation).
3. In my opinion, the introduction needs a bit of rewriting, as it is slightly confusing. A lot of previous work is cited, but it is difficult to position the paper inside the state-of-the-art. For instance, on page 2, it is mentioned in the beginning of the first paragraph that the previously reported magnet-free approaches to nonreciprocity exhibit strong limitations, while at the end of the very same paragraph it is written to some recent approaches overcome the aforementioned limitations. It is therefore extremely difficult for the reader to assess the novelty of the manuscript.
4. Page 10, when discussing their results, the authors explain that the proposed device could work better with a weak magnetic field. This is a disappointing conclusion for a manuscript aiming at designing a magnet-free optical insulator. Can the authors comment on that?

Authors' response to the Reviewers

Response to Reviewer #1

Reviewer #1 general Remarks to the Author:

Comment part 1:

The manuscript by Yuma Kawaguchi et al is interesting and based on convincing motivation. However, I have problems with finding sufficiently new results important for the field of physics. I have doubts whether the overall novelty of the presented research justifies publication in Nature Communications. The authors propose utilization of semiconductor transition metal dichalcogenides in design of devices with optically induced nonreciprocity. The idea is based on the phenomenon of optically induced circular dichroism. The authors present experimental results proving appearance of such dichroism in WS₂ samples.

Authors' response to Comment part 1:

We thank Reviewer#1 for finding our work “interesting” and “based on a convincing motivation”. We were surprised to see their comment that our work might not contain “new results important for the field of **physics**” as our work reports **photonics**-related results and, as we understand, the scope of Nature Communications does allow publishing research in this field. We hope that the reviewer will consider our work in the photonics context, and we hope that he/she will find the revised paper and our new results convincing.

Comment part 2:

However, the observation of such dichroism is not really new. Actually circular dichroism was studied for many years by means of pump-probe experiments including measurements of time resolved Faraday and Kerr rotation. The investigated systems were different including DMS-s layers, DMS quantum wells, doped semiconductor quantum wells and different semiconductor materials including TMDs. For certain examples the relaxation time was shown to be quite long. Therefore, in my opinion, the experimental data presented in the manuscript is neither new nor represents real experimental state of the art.

Authors' response to Comment part 2:

We thank Reviewer #1 for pointing out earlier studies of circular dichroism, many of which we are aware of. We should point out that the well-known inverse Faraday effect (photoinduced magnetization), which is described even in textbooks, also is a form of nonreciprocal photoinduced dichroism.

Reviewer #1 mentioned numerous materials where the photoinduced dichroism can emerge, and this is indeed the case, however the materials mentioned are not 2D materials and are not easy to integrate into existing photonic platforms in the same way as TMDs. Moreover, some of these materials are magnetic, i.e., exactly the type of materials we would like to avoid. We believe that we clearly explained our motivation in the introduction of the manuscript.

As for earlier work on the Kerr effect in TMDs, it was mainly aimed at lifting the valley degeneracy (Nature Materials, 14, 290 (2015)) and the valley-selective optical Stark effect considered relied on extremely high pump powers. Other reports by Hong-Kun Park group (Phys. Rev. Lett. 120, 037402 (2018)), K.F. Mak group (Nano Lett. 2018, 18, 3213–3220 (2018)), as well as the time-resolved Faraday and Kerr rotation experiments by other groups, were aimed at investigating pump-induced bistability and investigating the carrier dynamics in TMD materials, including magnetically induced dichroism we try to avoid. All these studies did not show or even aimed at any specific photonic application such as nonreciprocity and nonreciprocal devices.

Moreover, as the Reviewer correctly mentions, the earlier works primarily focus on the time-resolved response of the pump-probe kind. Our work, however, is not time-resolved pump-probe type and utilizes a long-lived dichroic response in TMD (with non-pulsed probe) to achieve nonreciprocal optical response, i.e. different reflection/transmission in the two opposite directions. Our work for the first time directly demonstrates the nonreciprocal response of TMDs by probing transmission/reflection in two opposite directions. We also note that, while long valley polarization times were studied before, no related long-lived dichroic response was demonstrated or used for studying nonreciprocal optical response.

To the best of our knowledge, there are currently no reports focused on using the photoinduced circular dichroic response of TMD monolayers for optical nonreciprocity or nonreciprocal integrated photonic device applications, including for optical isolation. Therefore, we are confident that our work has a significant novelty and will be of interest to the readership of Nature Communications.

Regarding the comment on our experiment not being at “state of the art” level, we are surprised by this opinion. Our original setup and the data were sufficient to demonstrate the dichroic response. Nonetheless, in order to address comments of the Reviewers

(including the Reviewer #1), we have completely rebuilt our experimental setup. The new setup allows us to directly observe nonreciprocal response for two opposite propagation directions. We hope that the Reviewer #1 will find our new experimental setup more “state of the art” and will consider that it is not the goal of our study to investigate carrier dynamics in TMDs, but to demonstrate that long-lived dichroic response in TMD monolayers can be used in photonic applications to engineer nonreciprocal optical responses all-optically without the need for external magnetic bias.

To be more specific, we have revised our experimental setup to allow for probing response of 2D materials in two opposite directions of propagation of light. The setup was used to show, for the first time (to the best of our knowledge), a nonreciprocal reflection of light from a TMD monolayer biased by circularly polarized pump. This result itself is very novel, as mentioned by the Reviewers #2 and #3.

Comment part 3:

The proposed simple theoretical model is phenomenological and gives no real insight into physics of exciton-exciton or exciton-carrier interactions. The microscopic mechanism of the long-lasting circular dichroism remains unknown.

Authors' response to Comment part 3:

We thank Reviewer #1 for sharing their opinion. However, it is not the goal of our work to study the microscopic origin of the long-living excitations in TMDs, which was investigated quite intensively in the past [Phys. Rev. B 90, 155449(2014), Nature Physics 11 830-834(2015), Nano Lett., 15, 5, 2794–2800(2015), Phys. Rev. Lett. 119, 137401(2017), Nano Lett., 19, 4083–4090(2019)]. A more detailed description of these prior works was added to the revised paper to make it more consistent and the corresponding references were cited.

For this reason, for our purpose of demonstrating nonreciprocity, it was sufficient to employ the model developed by Hong-Kun Park's group (Phys. Rev. Lett. 120, 037402 (2018)) extended by us to include valley polarization. This model was shown to describe the response of TMDs quite well and we do not see any reasons not to employ it as it can perfectly reproduce our experimental results. Therefore, we believe that our extension of the model (with valley polarization added) is fully adequate for our purpose of describing nonreciprocal behavior.

As of the “phenomenological” character of the model used, we would like to note that in the photonics community it is common practice to describe optical properties in terms of effective or phenomenological response parameters, such as the dielectric constant or the surface conductivity. These phenomenological parameters do not intend to describe microscopic light-matter interactions in materials, yet, when properly established, they

are sufficient to describe the macroscopic response and to design practical photonic devices.

Comment part 4:

In particular it is not clear whether the samples under examination are doped (p or n type) or the leading mechanism is related to dark exciton populations.

Authors' response to Comment part 4:

We thank the Reviewer 1 for raising this question. Indeed, our samples are doped, which, as suggested by other studies on valley polarization in TMDs, is indeed crucial for the long-lived dichroic response observed and utilized by us. The corresponding explanation and the references were added to the revised manuscript as follows:

“The fact that the dichroism does not vanish over such long integration times (compared to pump duration) indicates the presence of long-lived valley-polarized excitations in the system. Indeed, while the excitons in TMDs are known to have rather shorter lifetimes of less than 2 ps, recent time-resolved pump-probe experimental studies have suggested that the lifetime of photoexcited free carriers can be as long as few ns^{73–76} at room temperature, and may exceed values of 10 ns at cryogenic temperatures⁷⁷ and reach 130 ns if samples are electrically gated⁷⁸. Such long relaxation times was attributed to the presence of valley polarized resident carriers^{75–79}. Considering that we use a pulsed excitation, we expect that even stronger nonreciprocal dichroic response may be found for continuous wave pump, which can make our proposed approach to nonreciprocity even more viable for practical applications.”

Regarding dark excitons, they are unlikely to contribute in the observed dichroic response since i) the experiment was performed at room temperature [Nature Nanotechnology, 12, 856–860(2017)] and ii) the polarization of incident light is predominantly in-plane (dark excitons have out-of-plane polarizability [Phys. Rev. Lett. 119, 047401 (2017), 2D Materials 4, 021003 (2017)]).

Comment part 5:

The claim that the microscopic picture is illustrated in Fig. 1b is exaggerated. In my opinion any conclusions would require more careful analysis taking into account dynamics of different processes (relaxations and lifetimes of different excitonic complexes). Therefore, this part of the manuscript seems to be speculative and nebulous.

Authors' response to Comment part 5:

In the original submission, the schematic Fig. 1b was given solely with the aim of illustration to the broader photonics community, which might not be familiar with the excitations in TMDs. Thus, it was not intended to give a precise microscopic mechanism of observed optical response, which has been widely studied and understood by others [Phys. Rev. B 90, 155449(2014), Nature Physics 11 830-834(2015), Nano Lett., 15, 5, 2794–2800(2015), Phys. Rev. Lett. 119, 137401(2017), Nano Lett., 19, 4083–4090(2019)]. However, to avoid any “exaggerated” picture, in the revised version we removed this schematic and we have revised the text accordingly to describe and refer to the previous works on long-lived valley polarization in doped TMDs at room temperature.

We hope Reviewer #1 will take into consideration that in our work we solely aim at the demonstration of nonreciprocity and potential use of photoinduced dichroism for integrated photonic applications. Therefore, for our purpose it is sufficient to know that the valley polarization is preserved on longer timescale giving rise to long-lived dichroic response and nonreciprocity in TMD monolayers. We note that both pulsed and continuous-wave excitations were performed earlier in different contexts (Nano Lett. 2018, 18, 3213 (2018), Phys. Rev. Lett. 120, 037402 (2018)), but without any reference to nonreciprocity.

Comment part 6:

The main novelty of the paper resides in proposal of the design of optical isolator device. Unfortunately, the proposal is not supported by any experimental efforts which strongly limits its value.

Authors' response to Comment part 6:

We thank the Reviewer for their criticism. In the revised version of the manuscript, we present a new setup that allows us to directly demonstrate the free-space nonreciprocal response and isolation based on photoinduced nonreciprocity (see also additional results for nonreciprocal transmission in the Supplement).

Regarding the proposed theoretical concept of integrated devices based on a ring resonator, this geometry is proposed to boost the nonreciprocal response in a TMD monolayer to reach values relevant for practical photonic applications. We do not believe that having a theoretical result can be considered a problem precluding publication, especially considering that purely theoretical papers are regularly disseminated in Nature Communications. We also note that our proposed device operation concept is novel on its own and holds the promise for photonics applications by offering a highly desirable nonreciprocal response.

Comment part 7

In particular the time integrated results obtained in pump-probe experiment should be used with caution. In my opinion they cannot be used directly as a starting point for the device analysis. The operation of the real device might be strongly affected by temporal variation of optical properties. The correct analysis should be based on experiments with continuous wave excitation.

Authors' response to Comment part 6:

We thank Reviewer #1 for this comment. To further corroborate our conclusions and to address the criticism regarding continuous wave excitation, we performed additional experimental studies with a new source. While we do not have access to a continuous wave laser for the frequency close enough to the exciton resonance, we were able to perform measurements with a different supercontinuum laser with longer 190 ps pulse duration (vs 30 ps in previous studies) and the same repetition rate of 80 MHz (~12 ns pulse period). Our new results are consistent with our previous results reported in the original manuscript.

In addition, we would like to mention again that the long-lived valley polarization in doped TMDs is well known and literature refers to the valley-polarization lifetimes exceeding 100 ns [Phys. Rev. Lett. 119, 137401(2017), Nano Lett. 2019, 19, 4083–4090(2019)], which is even larger than our pulse period (12 ns). Therefore, considering these earlier studies and our own results, we conclude that the use of a continuous wave source will not change any conclusions made by us, but can potentially make the nonreciprocal response even stronger.

Thus, both, our experiments (in the current and previous submission) as well as studies by others fully support our claim of long-lived dichroic response. We emphasize again, that this is not a goal of our work to study carrier dynamics and physics of light-matter interactions in TMDs, which has been an active field of research on its own, but our goal is to demonstrate the nonreciprocal character of the dichroism and the possibility to use this response in practical photonic devices, which is unambiguously shown in our revised paper.

Reviewer #1 minor remarks

Additionally to my general objections I have a few minor remarks: The schematic drawings are not readable - the circular symbols representing circularly polarized light dominate everything making other elements less visible. Particularly the device design

in fig 4 is not clear. The abbreviations CW and CCW might be somehow misleading as CW laser usually means “continues wave” operation, not polarization.

Authors’ response to Reviewer #1 minor remarks

We thank the Reviewer for their constructive criticism of our graphics. In the revised manuscript all figures were revised, and the criticism was addressed. We hope that Reviewer #1 will find our new images more appealing and explanatory. As of CW and CCW notation, we are quite clear about these abbreviations, introducing them early in the text. We do not think it will cause any confusion since this notation is standard in our field of research. However, if the Reviewer still believes they may cause a confusion, we are open to redefining them.

Reviewer #1 summary

Taking into account all weakness I believe that the work in its present form should not be published in Nature Communications.

Authors’ response to Reviewer #1 summary

We thank Reviewer #1 for their thorough reading of the manuscript and for the suggested revisions. The manuscript was significantly revised, and, thanks to the Reviewers’ encouragement, the experimental setup was rebuilt, and new experimental studies were performed to provide a direct evidence of nonreciprocity. We hope that Reviewer #1 will find our revision satisfactory and the manuscript suitable for publication in Nature Communications.

Response to Reviewer #2

Reviewer 2 general remarks

Yuma Kawaguchi and co-authors experimentally demonstrate photoinduced nonreciprocity in transition metal dichalcogenides (TMDs) pumped by circularly polarized light. The authors claim that the nonreciprocity stems from the valley-selective exciton-exciton interactions that result in the nonlinear circular dichroism. Because the nonreciprocity is achieved without the involvement of an external magnetic field, it may be useful for future all-optical integrated photonics. As an example, they propose an optical isolator based on the optically biased TMD monolayer. The results are convincing, timely, and important to the field of nonreciprocal optics and 2D materials. Besides, the manuscript is very well-written, and the graphs are clear of understanding.

In terms of the quality of the text, the manuscript fulfils the standards of Nature Communications. Therefore, I would like to recommend publication of this work in Nature Communications.

Response to the General Remarks to the Author

We thank Reviewer #2 for their positive evaluation of our work and for noting the quality of the text and figures, which were further improved in the revision. We agree that all-optical nonreciprocity can indeed be revolutionary for photonics applications, especially considering the ease of integration of 2D materials into existing photonic platforms. We are grateful for the recommendation to publish our work in Nature Communications, and we also hope that Reviewer #2 will appreciate our new results, which now directly demonstrate photoinduced nonreciprocity by probing propagation in two opposite directions.

Reviewer #2 minor suggestions

Suggestion 1:

In the introduction, the authors state that the drawback of the conventional nonreciprocal optical devices based on magneto-optical materials is the bulky external magnetic bias. However, their approach proposed in this work relies on an external laser source for optical pumping. Is it also bulky?

Authors' response to Suggestion 1:

Unlike magnets, which are bulky, the pumping laser can be fully integrated on chip, which paves the way towards fully integrated nonreciprocal photonic systems. Indeed, the observed effect appears at rather moderate pump fluence. In our experiment we used 450 μW maximal pump beam focused on a 25 μm x 25 μm area, which, for the integrated setup in Fig. 4 with a TMD sample of size 3.3 μm by 50 nm implies a reduction of power by $\sim 3,000$ times, which corresponds to the power of 0.2 μW . This power can be further reduced by using an integrated scheme shown in Fig. R1, where we propose to use the pump field enhancement in the ring resonator itself. Such fully integrated setup has clear advantages over current magnetic nonreciprocal devices. It also offers tunability by switching on/of the pumps, effectively yielding an all-optical control of nonreciprocity. We clarified this point in the revised version of the manuscript.

Fig. R1. Schematic of all-optically tunable isolator with switchable isolation direction. Top and bottom schemes show cases of backward and forward isolation, respectively. Control of directionality is achieved by pumping at ports Pump 1 or Pump 2, which leads to the modes with opposite propagation direction and chiral near-field of opposite handedness in the ring resonator. No pump would yield reciprocal (bidirectional) regime.

Suggestion 2:

In the second paragraph of the introduction, the authors state that “However, this form of self-bias nonreciprocity comes with some drawbacks that hinder its widespread applicability.” Can the authors make it clear that what the drawbacks are?

Authors’ response to Suggestion 2:

Self-biasing nonreciprocity has a few drawbacks that have been pointed out in recent papers. First, the nonlinear effects can lead to the signal distortion, which will have a negative effect on the data transfer rates. In addition, self-biasing precludes a true form of isolation, as highlighted in the context of “dynamic reciprocity” suffered in these schemes [Nature Photonics 9, 388–392 (2015), Nature Photonics 9, 359–361 (2015)], therefore these types of nonreciprocal devices can be operated only for pulsed operation. We clarified these points in the revised version of the manuscript.

Suggestion 3:

In the experiments, the authors pump the TMDs with an intense circularly polarized laser radiation and illuminate the TMDs with a circularly polarized probe signal. They found that “the probe signals illuminating from opposite angles with opposite helicity, time-reversed versions of each other, will experience different reflectivity.” Can the

probe signals be replaced with linearly polarized light? What is the reflectivity expected to see? I think this is important because circular dichroism can also arise from the structure chirality.

Authors' response to Suggestion 3:

We thank Reviewer #2 for this suggestion. Our system is absolutely achiral and to confirm this (and also to exclude any possibility of the substrate induced bianisotropy/dichroism), we have rebuilt our setup to be able to measure the difference between forward and backward propagations under pumps of opposite handedness. We hope these new results, which, in fact show a simplest nonreciprocal device operation based on the observed dichroism, will convince Reviewer #2 in the correctness of our conclusions.

We also followed Reviewer #2's suggestion and performed measurements for a linearly polarized pump. As expected, no dichroism was observed. The results are given in Fig. R2 below and in the Supplement to the revised paper.

Fig. R2. Absence of the dichroic response under the linearly polarized pump.

Suggestion 4:

Below Eq. (2), it says "where $\sigma K = \sigma K(ICW)$ and $\sigma K' = \sigma K'(ICCW)$ are surface conductivities for the two valleys in the circularly polarized (CP) basis. The form of Eqs. (2) follows directly from the fact that the response of each valley is selective with

respect to the handedness of the optical field, and therefore it is described by the matrices $\hat{\sigma}KCPB=[\sigma K,0;0,0]$ and $\hat{\sigma}K'CPB=[0,0;0,\sigma K']$ in the circularly polarized basis (CBP).” The first “circularly polarized (CP) basis” should be “linearly polarized basis”? The second “circularly polarized basis (CBP)” should be “circularly polarized basis (CPB)”.

Authors’ response to Suggestion 4:

We thank Reviewer #2 for the comment. The confusion comes from the use of the same letters for the conductivity tensor in the linear basis and the scalar conductivities in the circular basis. In the revised text we either modified or elaborated our notations to avoid any confusion.

Suggestion 5:

On page 7, “as in Fig. 3a,b insets/zoom-ins” should be “as in Fig. 4a,b insets/zoom-ins.”

Authors’ response to Suggestion 5:

As per the Reviewer #2 suggestion the text was corrected.

Suggestion 6:

In Fig. 4, can the authors show the field patterns in the waveguide and ring resonator to better understand the operation of the optical isolator?

Authors’ response to Suggestion 6:

As per the Reviewer #2 suggestion the image with the field profile was added to the manuscript in the revised Fig. 4.

Response to Reviewer #3

Reviewer 3 General Remarks

This manuscript reports on the use of photo-excited 2D materials (WS_2) as the basis of a magnet-free non-reciprocal devices for photonics. After an introduction to remind the state-of-the-art, the manuscript consists of three main sections: (i) description and theoretical modeling of the proposed effect ; (ii) an experiment to demonstrate the existence of the effect ; and (iii) the design of a nonreciprocal device, an optical insulator (a study that remains purely theoretical at this stage).

The main idea of the paper is that a single layer of transition metal dichalcogenide, excited by an intense and circularly polarized pump at the excitonic resonance wavelength, will exhibit a polarization-dependent reflectivity to a probe beam (as

sketched in Fig. 1a). This claim sounds convincing, and an experiment is proposed to support it (Figure 2).

My concern (and main criticism) is that although this figure is entitled "Experimental demonstration of photo-induced nonreciprocal circular dichroism", I do not think that the claim of a nonreciprocal behavior is demonstrated at all in this experiment. From what is reported in the manuscript, I understand that the authors demonstrate photo-induced circular dichroism (a very interesting and important result by itself) but NOT photo-induced nonreciprocal circular dichroism.

Authors' response to General Remarks

First, we would like to thank Reviewer #3 for their thorough reading of the manuscript, for their inspiring comments and important suggestions, which, we believe, allowed us to significantly boost the quality of our work. As detailed in our point-by-point response given below, we have completely rebuilt our experimental setup to unequivocally demonstrate not only the dichroism, but also the nonreciprocal response of an optically biased TMD encapsulated in a symmetric optical environment to eliminate any possible substrate effects. We hope that Reviewer #3 will find our results convincing and worth of dissemination in Nature Communications.

Reviewer 3 Specific Remark 1

To demonstrate nonreciprocity, some kind of wavevector inversion should be involved, and the experimental setup (shown in the supplementary information) do not allow that. Contrary to what is sketched in Fig. 1a, the actual experiment does not involve any change of the parallel component of the probing beam's wavevector. The probe beam impinge the sample in normal incidence and the reflected light is collected through the same objective lens. I fail to see how such an experiment could demonstrate anything about reciprocity.

Authors' response to the Specific Remark 1

We thank Reviewer #3 for pointing out the important differences between the schematics of the original paper and the experimental setup, which may lead to some misunderstanding of the reported results. Indeed, as the Reviewer suggests, the dichroic response was previously reported for metasurfaces, attributed either to the substrate effect or in-plane asymmetry of metamolecules, or both. All these effects can be attributed to the effective bianisotropic response (or nonlocality), which, while giving rise to circular birefringence and dichroism, are not nonreciprocal in nature.

Fig. R3. Schematic of microscope set up in wave vector inversion geometry for optical Nonreciprocity measurements. LP: linear polarizer, LPF: Long pass filter, TL: Tube lens, L: lens, M: Mirror, FM: Flippable mirror, BS: 50/50 beam splitter.

Thus, Reviewer #3 is absolutely correct that the experimental studies could be more convincing if performed in a specific manner with the inversion of the propagation direction. Therefore, inspired by Reviewer #3's comments, we have completely redesigned our experiment and we have rebuilt our experimental setup as shown in Fig. R3 to probe the nonreciprocal response with respect to the probe beam wavevector inversion while keeping fixed the pump direction and helicity (can be switched between LCP and RCP). As it can be seen, the beam now can come from the two opposite sides and (for the case of this symmetric structure) the nonreciprocity results in different transmission/reflection for light incident from two opposite direction.

Fig. R4 shows the nonreciprocal responses (reflection) measured for the case of fixed one-handed pump and two probes impinging from the two opposite (k_x and $-k_x$) directions. We believe that these new results unambiguously confirm our claim of photoinduced nonreciprocity and this simple setup clearly shows a nonreciprocal optical response induced by the circularly polarized pump. More detailed measurements data are provided in the revised supplementary information.

Fig. R4. Experimental demonstration of photoinduced nonreciprocity due to dichroic response in TMD in wavevector inversion setup.

Reviewer 3 Specific Remark 2

If I understand correctly, the point made by the authors is as the sample is a 2D material, then circular dichroism must be nonreciprocal, as stated page 3 of the manuscript: "It is worth noting here that the circular dichroic response of any planar 2D systems, including 2D materials, is necessarily nonreciprocal, since reciprocal circular dichroism, known as optical activity, requires nonlocal bianisotropic response, which is possible only in structures with finite thickness". This claim is supported by two bibliographical references.

I had a look at these papers:

- reference 57 does not evoke reciprocity at all;
- reference 56 does evoke reciprocity but does not provide direct evidence to the assumption made by the authors.

As a consequence, I do not think this assumption is well-supported, and a more convincing demonstration must be provided.

Authors' response to Specific Remark 2

We thank the Reviewer for pointing out this possible ambiguity. Indeed, this sentence is correct only in the case of no substrate and no structuring, which can give rise to effective bianisotropy and intrinsic or extrinsic chirality of the structure. We originally assumed that both mechanisms of optical activity are excluded in our case since the substrate refractive index (1.5 for the BK7 glass) was relatively close to the one of air and the TMD monolayer is homogenous on the atomic scale. However, we agree that the substrate still could play some role and that in the previous asymmetric cladding the effect could still originate from asymmetry of the structure. A combination of loss and the presence of the substrate therefore could probably lead to some form of dichroism. We removed this ambiguous claim in the revised manuscript. Also, as described below, the experiment was redone for fully symmetric cladding.

Reviewer 3 Specific Remark 3

Furthermore, there might be another problem. Even if the above assumption is correct, the actual sample studied in the manuscript is not 2D: it is a 2D layer sitting on a substrate. The potential role played by the substrate is completely eluded in the whole manuscript. Researchers working in the field of chiral plasmonics have shown, when working with planar metallic nanostructures, the critical role played by the substrate. This question actually led to a strong debate in the community - see, e.g. Kuwata-Gonokami, M. et al. Giant Optical Activity in Quasi-Two-Dimensional Planar Nanostructures. Phys. Rev. Lett. 95, 735 (2005); Collins, J. T. et al. Chirality and Chiroptical Effects in Metal Nanostructures: Fundamentals and Current Trends. Advanced Optical Materials 5, 1700182 (2017); Drezet, A. & Genet, C. Reciprocity and optical chirality. arXiv.org physics.optics (2017).

It is now generally admitted that "planar chirality" is associated to the symmetry-breaking brought by the substrate.

I cannot say if such effects are involved here, but the question must be discussed in the manuscript. As a relevant example, the authors write on page 4 of the manuscript: "It is worth highlighting that here the notations of CW and CCW specify the handedness of the electric field rotation in the TMD plane, irrelevant of the propagation direction". Does this assumption still hold if there is a substrate? I would say that the handedness is not the same looking from the air side or from the substrate side. It is worth noting that the directions from which the sample is pumped and probed are not specified.

Authors' response to the specific Remark 3

We thank Reviewer #3 for pointing out the possible role of the substrate. To address these concerns, a new sample was prepared with a TMD monolayer embedded into a completely symmetric optical environment, which, combined with the new experimental

setup (also fully symmetrized), should eliminate any possible doubts about nonreciprocity of the measured dichroic response. We believe that this new sample, along with the new experimental setup capable of probing the response from two opposite directions, eliminates any possible doubt about nonreciprocal response of the structure under circularly polarized pump bias.

Specifically, to exclude any possibility of the substrate-induced dichroism we have fabricated a new sample with structure shown in Fig. R5. A WS₂ TMD monolayer (sandwiched between two hBN layers) was transferred to a BK7 glass cover slip (n=1.5) and coated with 495PMMA A11 photoresist of the same refractive index (n=1.5) as the glass cover slip. We then measured its nonreciprocal dichroic response with the use of our rebuilt set up shown in revised Fig. 1e (and above as Fig. R3) and we observed pump-induced nonreciprocity, as plotted in Fig. R4. We hope these new results will resolve any concerns raised by the Reviewer #3.

Fig. R5. Schematic of the WS₂ monolayer encapsulated between two thin hBN layers and symmetrically sandwiched between materials (BK7 glass cover slip and thick PMMA slab) of the same refractive index. Red: WS₂ monolayer; light blue: hBN layers.

Other comments/questions:

Reviewer #3 minor suggestions

Suggestion 1:

1. The schematic of the experimental set-up should be moved from the supplementary information to the main text. It is of critical importance to understand the experiment.

Authors' response to Suggestion 1:

As per the reviewer suggestion, we have included our revised experimental setup as a schematic in the revised manuscript Fig. 1e.

Suggestion 2:

2. On page 3, the authors write: "reciprocal circular dichroism, known as optical activity"; I am not sure to understand correctly. Optical activity refers to any chiroptical effect involving a different response to right- and left-circularly polarized light (see L.D. Barron,

Molecular light scattering and optical activity, Cambridge Univ. Press, page 1). Hence, optical activity includes circular dichroism (as well as optical rotation).

Authors' response to Suggestion 2:

We agree with the Reviewer that any chiroptical response can be referred to as optical activity. We corrected the text accordingly to avoid any possible confusion.

Suggestion 3:

3. In my opinion, the introduction needs a bit of rewriting, as it is slightly confusing. A lot of previous work is cited, but it is difficult to position the paper inside the state-of-the-art. For instance, on page 2, it is mentioned in the beginning of the first paragraph that the previously reported magnet-free approaches to nonreciprocity exhibit strong limitations, while at the end of the very same paragraph it is written to some recent approaches overcome the aforementioned limitations. It is therefore extremely difficult for the reader to assess the novelty of the manuscript.

Authors' response to Suggestion 3:

As per the reviewer suggestion we have revised the introduction to make it clearer and more coherent.

Suggestion 3:

4. Page 10, when discussing their results, the authors explain that the proposed device could work better with a weak magnetic field. This is a disappointing conclusion for a manuscript aiming at designing a magnet-free optical insulator. Can the authors comment on that?

Authors' response to Suggestion 3:

We understand the Reviewer's frustration about our suggestion to improve the device operation with a weak magnetic bias. This suggestion was assuming that the magnetic field would be still much weaker than that in common magneto-optical devices so that only a tiny magnet (potentially integrable into the device) would work.

However, our new results obtained with the sample with better valley polarization stability (due to hBN encapsulation from both sides) show that this is not even necessary. Thus, we see that one can achieve a strong nonreciprocal response even with the dichroic response we observe at room temperature. Magnetic field is not really needed. Therefore, the corresponding suggestion was removed.

Reviewer #3 general remark (conclusion)

To conclude, this manuscript reports on a very interesting experiment on photo-induced circular dichroism. This result most certainly deserves publication and it might be appropriate for Nature Communications. However, in my opinion publication at this stage would be premature. Either further experiments are required to demonstrate the nonreciprocal behavior, or the proposed interpretation is not clearly exposed: in any case a substantial rewriting appears necessary to clarify the two issues I mentioned above.

Authors' response to the general remark (conclusion)

We thank the Reviewer #3 for noting that our experiment is “*very interesting*” and that our work “*most certainly deserves publication*”, and “*might be appropriate for Nature Communications*”. As suggested by the Reviewer #3, the “*farther experiments*” were performed to confirm the true nonreciprocal character of the observed dichroism. The figures and the text were revised accordingly. We hope that our new results obtained for a sample with a symmetric cladding and in a completely new setup, both inspired by the Reviewer #3 comments, will address their concerns.

In particular, a symmetric cladding eliminates the role of substrate, which addresses the “*issue*” #2 raised by the Reviewer #3, while the new setup and a direct demonstration of nonreciprocity, should fully address the “*issue*” #1 about the nonreciprocal character of the observed dichroism. We hope Reviewer #3 will find our results convincing and our revised manuscript suitable for publication in Nature Communications.

REVIEWERS' COMMENTS

Reviewer #1 (Remarks to the Author):

I appreciate the response given by the authors and the efforts they did in order to convincingly demonstrate nonreciprocity of the observed photoinduced circular dichroism. The measurements are now performed for two directions of the wave propagation, which gives no surprises but is probably sufficient. Nevertheless I am sorry to say that my general opinion on the overall novelty and maturity of the presented research is not changed. The photo induced circular dichroism is very well-known effect. This is also admitted by authors in their response. Moreover the presented motivation to avoid necessity to use external magnetic field is not equivalent to exclusion of magnetic materials or materials containing magnetic ions. Despite amendments of the experiment the section devoted to design of the optical isolator is still purely theoretical. Therefore the experimental novelty is restricted to observation of the known effect for new kind of material at room temperature. It is clearly presented that the observed circular response lasts long at room temperature. This finding is interesting. But, even when recognizing importance of this result, one have to notice that the research is not sufficiently elaborated. The mechanisms, limiting factors and dynamics of the observed dichroism are not sufficiently explored. But, if the mechanism of the effect is out of the scope of the present work, then in my opinion the goal should be focused on demonstration of preliminary version of the proposed device, preferably with continue wave excitation. Unfortunately, this is not a case of the current manuscript. Therefore I hold my previous opinion that the work in its present form should not be published in Nature Communications.

Reviewer #2 (Remarks to the Author):

The authors have provided satisfactory answers to all questions raised by three referees. After revision, the manuscript has been improved substantially. Therefore, in my opinion, this revised version is ready for publication in Nature Communications.

Reviewer #3 (Remarks to the Author):

The manuscript from Guddula and coworkers is the revised version of a manuscript I previously reviewed. The original manuscript aimed at demonstrating photo-induced optical non-reciprocity in a layer of WS₂. In my previous review, I was concerned by two caveats which, I thought, flawed the demonstration: (i) the fact that the original experiment did not involve any change in the propagation direction ; and (ii) a potential role of the substrate. Answering these issues was a challenging task, implying to design a new experiment and a new sample, as well as performing a whole run of complex measurements. I am really glad to see that the authors took the time required to design and perform these experiments. But the most important point is that the experiments beautifully worked.

The new experiment, reported in the present manuscript, allows the authors to change the probe's propagation direction with respect to the sample and the pump beam. Also, to avoid any unwanted effects coming from the substrate, the sample was designed in a symmetrical way (as a sandwich). The results, shown in Fig. 2 and 3, ambiguously and clearly demonstrate the author's claim for pump-induced non-reciprocity. A comprehensive set of data covering all possible configurations is provided in the Supplementary Information. A control experiment using linear polarization is also provided. It is solid, neat experimental work. To the best of my knowledge, it is the first demonstration of such an effect in a TMD layer.

Besides, the authors answered clearly to all my minor comments, and I think that the manuscript is now free from any ambiguity. Based on that, I strongly recommend the publication of this work in Nature

Communications. The manuscript could be published as it.

I have only a minor comment, actually a mere matter of curiosity. The authors performed the non-reciprocity measurements using the "sandwiched" sample (in order to remove any potential effect induced by substrate asymmetry). But did the authors also check what happen with an asymmetric sample, i.e. with the TMD layer sitting on a substrate? If yes, did they observe any difference? If data is available, it would be a nice addition to the Supplementary Information.

AUTHORS' RESPONSE TO REVIEWERS' COMMENTS

Reviewer #1

Reviewer #1, general comment 1

I appreciate the response given by the authors and the efforts they did in order to convincingly demonstrate nonreciprocity of the observed photoinduced circular dichroism. The measurements are now performed for two directions of the wave propagation, which gives no surprises but is probably sufficient.

Authors' response to general comment 1

We thank the Reviewer for agreeing about the nonreciprocal nature of the observed phenomena and for noting that the measurement performed for two directions is sufficient to support this claim.

Reviewer #1, general comment 2

Nevertheless I am sorry to say that my general opinion on the overall novelty and maturity of the presented research is not changed. The photo induced circular dichroism is very well-known effect. This is also admitted by authors in their response. Moreover the presented motivation to avoid necessity to use external magnetic field is not equivalent to exclusion of magnetic materials or materials containing magnetic ions. Despite amendments of the experiment the section devoted to design of the optical isolator is still purely theoretical. Therefore the experimental novelty is restricted to observation of the known effect for new kind of material at room temperature. It is clearly presented that the observed circular response lasts long at room temperature. This finding is interesting. But, even when recognizing importance of this result, one have to notice that the research is not sufficiently elaborated. The mechanisms, limiting factors and dynamics of the observed dichroism are not sufficiently explored. But, if the mechanism of the effect is out of the scope of the present work, then in my opinion the goal should be focused on demonstration of preliminary version of the proposed device, preferably with continue wave excitation. Unfortunately, this is not a case of the current manuscript. Therefore I hold my previous opinion that the work in its present form should not be published in Nature Communications.

Authors' response to general comment 2

We thank Reviewer 2 for sharing their opinion about our observed phenomenon. We kindly disagree with their assessment of novelty of our work and results since we believe the paper should be evaluated as a whole and not as separate sections on dichroic response and the device design proposal. In addition, as we mentioned in our previous response, the mechanism of the observed long-lived room temperature response is still debated in the literature and is a subject of an active investigation by several groups. More important argument in support of the novelty of our work lays in the fact that this is first observation of the photoinduced dichroism in TMDs and due to the valley polarization, which is rather unique property of these 2D materials. We also note that this novel aspect is also emphasized by other reviewers.

To further emphasize the difference of the observed photoinduced dichroism in the TMD monolayer with that in other materials, in the revised paper we now outlined the most reasonable (from the standpoint of the current state of the field) explanation of the mechanism behind the observed phenomenon.

We believe, and this is corroborated by other reviewers' reports, that the observed effect has never been observed in TMD monolayers and can indeed be of significant interest for photonics community due to its magnitude even at room temperature, which makes it very promising for photonics application. We are also confident that this work will not go unnoticed by the materials sciences community and it will reignite research on photoinduced dichroism in different 2D materials with valley degree of freedom, which will help to reveal and to better understand the mechanism behind it.

Reviewer #2

Reviewer #2 general remark

The authors have provided satisfactory answers to all questions raised by three referees. After revision, the manuscript has been improved substantially. Therefore, in my opinion, this revised version is ready for publication in Nature Communications.

Authors' response

We appreciate Reviewer #2 remarks that the manuscript was improved substantially after the revision and for recommending our work for publication in its present form. We note that this revisions and improvements were largely inspired by Reviewers' comments and criticism, for which we are truly grateful.

Reviewer #3

Reviewer 3 general remarks

The manuscript from Guddula and coworkers is the revised version of a manuscript I previously reviewed. The original manuscript aimed at demonstrating photo-induced optical non-reciprocity in a layer of WS₂. In my previous review, I was concerned by two caveats which, I thought, flawed the demonstration: (i) the fact that the original experiment did not involve any change in the propagation direction ; and (ii) a potential role of the substrate. Answering these issues was a challenging task, implying to design a new experiment and a new sample, as well as performing a whole run of complex measurements. I am really glad to see that the authors took the time required to design and perform these experiments. But the most important point is that the experiments beautifully worked.

The new experiment, reported in the present manuscript, allows the authors to change the probe's propagation direction with respect to the sample and the pump beam. Also, to avoid any unwanted effects coming from the substrate, the sample was designed in a symmetrical way (as a sandwich).

The results, shown in Fig. 2 and 3, ambiguously and clearly demonstrate the author's claim for pump-induced non-reciprocity. A comprehensive set of data covering all possible configurations is provided in the Supplementary Information. A control experiment using linear polarization is also provided. It is solid, neat experimental work. To the best of my knowledge, it is the first demonstration of such an effect in a TMD layer.

Besides, the authors answered clearly to all my minor comments, and I think that the manuscript is now free from any ambiguity. Based on that, I strongly recommend the publication of this work in Nature Communications. The manuscript could be published as it.

Authors' response to Reviewer #3 general remark

We would like to thank Reviewer #3 for their very positive evaluation of our revised work and for their very encouraging and helpful comments in the previous review round, which inspired this substantial revision. We also thank the reviewer for recommending our work for publication in its present form.

Reviewer 3 minor comment

I have only a minor comment, actually a mere matter of curiosity. The authors performed the non-reciprocity measurements using the "sandwiched" sample (in order to remove any potential effect induced by substrate asymmetry). But did the authors also check what happen with an asymmetric sample, i.e. with the TMD layer sitting on a substrate? If yes, did they observe any difference? If data is available, it would be a nice addition to the Supplementary Information.

Authors' response

We appreciate Reviewer #3 comment. In our experiments we did not evaluate whether the effect would be suppressed or enhanced by the substrate or cladding. While in the previous submission we had somewhat asymmetric cladding, still, the contrast between the substrate (glass) and the superstrate (air) was probably too small to observe any substrate effects.

Although we did observe some enhancement in the dichroic response in the new samples (when compared to the old samples used in the previous submission), it could be solely attributed to the fact that in the second round of measurement the TMD samples were fully encapsulated by two hBN layers, which is known to yield a longer valley polarization lifetime. This alone could give rise to the stronger dichroic response. For this reason, the data available is no sufficient to make any reasonable conclusions about the substrate effect.

Therefore, to understand the role of substrate one would need to perform a whole new set of experiments for identical (e.g., fully encapsulated) TMD monolayers placed in different dielectric environments and probed in an identical optical setup (our setup was modified for the revision). As we are currently working on experimental realization of our idea of an integrated optical isolator, we will eventually be able to answer this question. As of now, however, we do not have any reasonable data and, therefore, we cannot make any conclusions regarding the substrate effects. Nonetheless, we

thank Reviewer #3 for this comment and we will investigate and report on the effect of the substrate in our future publications.